 

# Structural basis for molecular assembly of fucoxanthin chlorophyll *a*/*c*-binding proteins in a diatom photosystem I supercomplex

Koji Kato[1†], Yoshiki Nakajima[1†], Jian Xing[2†], Minoru Kumazawa[2], Haruya Ogawa[1], Jian-Ren Shen[1*], Kentaro Ifuku[2*], Ryo Nagao[3*]

[1]Research Institute for Interdisciplinary Science and Graduate School of Environmental, Life, Natural Science and Technology, Okayama University, Okayama, Japan; [2]Graduate School of Agriculture, Kyoto University, Kyoto, Japan; [3]Faculty of Agriculture, Shizuoka University, Shizuoka, Japan

**\*For correspondence:**
shen@cc.okayama-u.ac.jp (J-RS);
ifuku.kentaro.2m@kyoto-u.ac.jp (KI);
nagao.ryo@shizuoka.ac.jp (RN)

[†]These authors contributed equally to this work

**Competing interest:** The authors declare that no competing interests exist.

## eLife Assessment

This **important** study presents a high-resolution cryoEM structure of the supercomplex between photosystem I (PSI) and fucoxanthin chlorophyll *a*/*c*-binding proteins (FCPs) from the model diatom *Thalassiosira pseudonana* CCMP1335, revealing subunits, protein:protein interactions and pigments not previously seen in other diatoms or red/green photosynthetic lineages. Combining structural, sequence and phylogenetic analyses, the authors provide **convincing** evidence of conserved motifs crucial for the binding of FCPIs, accompanied by interesting speculation about the mechanisms governing the assembly of PSI-FCPI supercomplexes in diatoms and their implications for related PSI-LHCI supercomplexes in plants. The findings set the stage for functional experiments that will further advance the fields of photosynthesis, bioenergy, ocean biogeochemistry and evolutionary relationships between photosynthetic organisms.

**Abstract** Photosynthetic organisms exhibit remarkable diversity in their light-harvesting complexes (LHCs). LHCs are associated with photosystem I (PSI), forming a PSI-LHCI supercomplex. The number of LHCI subunits, along with their protein sequences and pigment compositions, has been found to differ greatly among the PSI-LHCI structures. However, the mechanisms by which LHCIs recognize their specific binding sites within the PSI core remain unclear. In this study, we determined the cryo-electron microscopy structure of a PSI supercomplex incorporating fucoxanthin chlorophyll *a*/*c*-binding proteins (FCPs), designated as PSI-FCPI, isolated from the diatom *Thalassiosira pseudonana* CCMP1335. Structural analysis of PSI-FCPI revealed five FCPI subunits associated with a PSI monomer; these subunits were identified as RedCAP, Lhcr3, Lhcq10, Lhcf10, and Lhcq8. Through structural and sequence analyses, we identified specific protein–protein interactions at the interfaces between FCPI and PSI subunits, as well as among FCPI subunits themselves. Comparative structural analyses of PSI-FCPI supercomplexes, combined with phylogenetic analysis of FCPs from *T. pseudonana* and the diatom *Chaetoceros gracilis*, underscore the evolutionary conservation of protein motifs crucial for the selective binding of individual FCPI subunits. These findings provide significant insights into the molecular mechanisms underlying the assembly and selective binding of FCPIs in diatoms.

## Introduction

Oxygenic photosynthesis in cyanobacteria, algae, and land plants converts solar energy into chemical energy and releases molecular oxygen into the atmosphere (*Blankenship, 2021*). The conversion of light energy takes place within two multi-subunit membrane protein complexes, known as photosystem I (PSI) and photosystem II (PSII), which perform light harvesting, charge separation, and electron transfer reactions (*Golbeck, 1992*; *Brettel and Leibl, 2001*; *Shen, 2015*; *Shevela et al., 2023*). To optimize light energy capture, numerous light-harvesting antenna subunits are associated with the periphery of the PSI and PSII core complexes, transferring excitation energy to the respective photosystem cores (*Blankenship, 2021*). These light-harvesting antennae exhibit significant diversity among photosynthetic organisms, both in protein sequences and pigment compositions, and can be broadly categorized into two major groups: membrane proteins and water-soluble proteins (*Blankenship, 2021*).

The membrane protein category primarily consists of the light-harvesting complex (LHC) protein superfamily (*Engelken et al., 2010*; *Sturm et al., 2013*), which absorbs light energy through chlorophylls (Chls) and carotenoids (Cars). The number and types of Chls and Cars vary significantly among LHCs, which can be grouped into green and red lineages, leading to color diversity in photosynthetic organisms (*Falkowski et al., 2004*). The green lineage includes green algae and land plants, while the red lineage encompasses red algae, diatoms, haptophytes, cryptophytes, and dinoflagellates (*Falkowski et al., 2004*). LHCs specific to PSI (LHCIs) bind to a eukaryotic PSI monomer, forming a PSI-LHCI supercomplex (*Hippler and Nelson, 2021*; *Shen, 2022*), the structures of which have been revealed by cryo-electron microscopy (cryo-EM) in various eukaryotes (*Hippler and Nelson, 2021*; *Shen, 2022*). In the red lineage, the number of LHCIs and their protein sequences and pigment compositions exhibit considerable variation among the PSI-LHCI structures of red algae (*Pi et al., 2018*; *Antoshvili et al., 2019*; *You et al., 2023*; *Kato et al., 2024*), a diatom (*Nagao et al., 2020a*; *Xu et al., 2020*), a cryptophyte (*Zhao et al., 2023*), and dinoflagellates (*Li et al., 2024*; *Zhao et al., 2024*).

Recently, we demonstrated the conservation and diversity of LHCIs among red-lineage algae through structural and phylogenetic analyses of PSI-LHCI supercomplexes (*Kato et al., 2024*). This study revealed that while the binding sites of LHCIs to PSI were conserved to some extent among red-lineage algae, their evolutionary relationships were weak. It is known that LHCIs have similar overall protein structures across photosynthetic organisms, with particular similarity in their three-transmembrane helices, regardless of whether they belong to the green or red lineages (*Hippler and Nelson, 2021*; *Shen, 2022*). However, individual LHCIs have altered their sequences and structures to adapt their respective binding sites to the PSI cores during the assembly of PSI-LHCI supercomplexes. These observations raise a critical question: how do LHCIs recognize their binding sites in the PSI core?

Diatoms are among the most essential phytoplankton in aquatic environments, playing a crucial role in the global carbon cycle, supporting marine food webs, and contributing significantly to nutrient cycling, thus ensuring the health and sustainability of marine ecosystems (*Field et al., 1998*). Diatoms possess unique LHCs known as fucoxanthin Chl *a/c*-binding proteins (FCPs), which differ in pigment composition and amino acid sequences from the LHCs of land plants (*Green and Durnford, 1996*; *Büchel, 2020*; *Wang and Shen, 2021*). Previous studies have reported the isolation and structural characterization of PSI-FCPI supercomplexes from the diatom *Chaetoceros gracilis* (*Nagao et al., 2020a*; *Xu et al., 2020*; *Ikeda et al., 2008*; *Nagao et al., 2019a*; *Nagao et al., 2019b*; *Nagao et al., 2019c*; *Nagao et al., 2019d*; *Nagao et al., 2020c*). Kumazawa et al. showed significant diversity in FCPs between *C. gracilis* and *Thalassiosira pseudonana*, with 46 and 44 FCPs identified, respectively (*Kumazawa et al., 2022*). These FCPs are categorized into multiple, closely related subgroups (*Kumazawa et al., 2022*), and their amino acid sequences are not entirely identical between the two diatoms. Consequently, comparing FCPIs, including their amino acid residues and protein structures at similar binding sites in PSI-FCPIs, may provide molecular insights into how FCPIs interact with PSI. However, an overall structure of the *T. pseudonana* PSI-FCPI supercomplex has yet to be solved.

In this study, we solved the structure of the PSI-FCPI supercomplex from *T. pseudonana* CCMP1335 at a resolution of 2.30 Å by cryo-EM single-particle analysis. The structure reveals a PSI-monomer core and five FCPI subunits. Structural and sequence comparisons highlight unique protein–protein

interactions between each FCPI subunit and PSI. Based on these findings, we discuss the molecular assembly and selective binding mechanisms of FCPI subunits in diatom species.

## Results and discussion
### Overall structure of the *T. pseudonana* PSI-FCPI supercomplex

The PSI-FCPI supercomplexes were purified from the diatom *T. pseudonana* CCMP1335 and analyzed by biochemical and spectroscopic techniques (*Figure 1—figure supplement 1*). Notably, the protein bands of PSI-FCPI closely resembled those reported in a previous study (*Ikeda et al., 2013*). Cryo-EM images of the PSI-FCPI supercomplex were obtained using a JEOL CRYO ARM 300 electron microscope operated at 300 kV. The final cryo-EM map was determined at a resolution of 2.30 Å with a C1 symmetry (*Figure 1—figure supplements 2 and 3*, and *Table 1*), based on the 'gold standard' Fourier shell correlation (FSC) = 0.143 criterion (*Figure 1—figure supplement 3A*).

The atomic model of PSI-FCPI was built based on the cryo-EM map obtained (see Methods; *Figure 1—figure supplement 3* and *Tables 1–3*). The structure reveals a monomeric PSI core associated with five FCPI subunits (*Figure 1A, B*). The five FCPI subunits were named FCPI-1–5 (*Figure 1A*), following the nomenclature of LHCI subunits in the PSI-LHCI structure of *Cyanidium caldarium* RK-1 (NIES-2137) (*Kato et al., 2024*). Specifically, the positions of FCPI-1 and FCPI-2 in the *T. pseudonana* PSI-FCPI structure (*Figure 1A*) correspond to those of LHCI-1 and LHCI-2 in the *C. caldarium* PSI-LHCI structure. The PSI core comprises 94 Chls *a*, 18 *β*-carotenes (BCRs), 1 zeaxanthin (ZXT), 3 [4Fe-4S] clusters, 2 phylloquinones, and 6 lipid molecules, whereas the 5 FCPI subunits include 45 Chls *a*, 7 Chls *c*, 2 BCRs, 15 fucoxanthins (Fxs), 7 diadinoxanthins (Ddxs), and 3 lipid molecules (*Table 3*).

### Structure of the *T. pseudonana* PSI core

The PSI core contains 12 subunits, 11 of which are identified as PsaA, PsaB, PsaC, PsaD, PsaE, PsaF, PsaI, PsaJ, PsaL, PsaM, and Psa29 (*Figure 1B*). The remaining subunit could not be assigned due to insufficient map resolution and was therefore modeled as polyalanines (*Figure 1—figure supplement 4A*). This unidentified subunit, designated as Unknown, occupies the same

**Table 1.** Cryo-electron microscopy (cryo-EM) data collection and structural analysis statistics.

| Complex | PSI-FCPI |
|---|---|
| PDB ID | 8XLS |
| EMDB ID | EMD-38457 |
| Data collection and processing | |
| Magnification | 60,000 |
| Voltage (kV) | 300 |
| Electron exposure (e⁻/Å) | 50 |
| Defocus range (μm) | −1.8 to −1.2 |
| Pixel size (Å) | 0.752 |
| Symmetry imposed | C1 |
| Initial particle images (no.) | 2,733,572 |
| Final particle images (no.) | 75,667 |
| Map resolution (Å) | 2.30 |
| FSC threshold | 0.143 |
| Refinement | |
| Initial model used | *De novo* model building |
| Model resolution (Å) | 2.25 |
| FSC threshold | 0.5 |
| Map sharpening *B* factor (Å²) | −36.0 |
| Model composition | |
| Non-hydrogen atoms | 37,640 |
| Protein residues | 3129 |
| Ligand molecules | 372 |
| Water molecules | 922 |
| *B* factors (Å²) | |
| Protein | 59.2 |
| Ligand | 71.9 |
| Water | 54.7 |
| R.m.s. deviations | |
| Bond lengths (Å) | 0.025 |
| Bond angles (°) | 2.46 |
| Validation | |
| MolProbity score | 1.98 |
| Clashscore | 11.8 |
| Poor rotamers (%) | 3.04 |
| EMRinger score | 5.70 |
| Ramachandran plot | |
| Favored (%) | 97.90 |

*Table 1 continued on next page*

**Table 1 continued**

| Complex | PSI-FCPI |
| --- | --- |
| Allowed (%) | 2.07 |
| Disallowed (%) | 0.03 |

site as Psa28 in the *C. gracilis* PSI-FCPI (*Nagao et al., 2020a*). The structural comparison reveals that Unknown closely resembles Psa28 in the *C. gracilis* PSI-FCPI (*Figure 1—figure supplement 4B*). Psa28, a novel subunit identified in the *C. gracilis* PSI-FCPI structure (*Nagao et al., 2020a*), follows the previously established nomenclature rule (*Kashino et al., 2002*). Historically, genes encoding PSI proteins have been designated as *psaA*, *psaB*, and so forth. PsaZ was identified in the PSI cores of *Gloeobacter violaceus* PCC 7421 (*Inoue et al., 2004*; *Kato et al., 2022*). Subsequent discoveries led to the designation of a new subunit as Psa27, which was identified in the PSI cores of *Acaryochloris marina* MBIC11017 (*Tomo et al., 2008*; *Hamaguchi et al., 2021*; *Xu et al., 2021*). Consequently, we designated this novel subunit as Psa28 (*Nagao et al., 2020a*). However, Xu et al. referred to this subunit as PsaR in the PSI-FCPI structure of *C. gracilis* (*Xu et al., 2020*).

Psa29 is newly identified in the *T. pseudonana* PSI-FCPI structure using ModelAngelo (*Jamali et al., 2024*) and the NCBI database (https://www.ncbi.nlm.nih.gov/) (*Figure 2*). The subunit corresponding to Psa29 was also observed previously in the *C. gracilis* PSI-FCPI structures (*Nagao et al., 2020a*; *Xu et al., 2020*), where it was modeled as polyalanines and referred to as either Unknown1 (*Nagao et al., 2020a*) or PsaS (*Xu et al., 2020*). Psa29 exhibits a unique structure distinct from the other PSI subunits in the *T. pseudonana* PSI-FCPI (*Figure 2A* and *Figure 1—figure supplement 4C*) and engages in multiple interactions with PsaB, PsaC, PsaD, and PsaL at distances of 2.5–3.2 Å (*Figure 2B–G*). Sequence analyses suggest that Psa29 has undergone evolutionary divergence between Bacillariophyceae (diatoms) and Bolidophyceae, the latter of which is a sister group of diatoms within Stramenopiles (*Figure 2H*), although this subunit has not been found in other organisms. The arrangement of PSI subunits in the *T. pseudonana* PSI-FCPI is virtually identical to that in the *C. gracilis* PSI-FCPI structures already reported (*Nagao et al., 2020a*; *Xu et al., 2020*). However, the functional and physiological roles of Psa29 remain unclear at present. It is evident that Psa29 does not have any pigments, quinones, or metal complexes, suggesting no contribution of Psa29 to electron transfer reactions within PSI. Further mutagenesis studies will be necessary to investigate the role of Psa29 in diatom photosynthesis.

The number and arrangement of Chls and Cars within the PSI core in the *T. pseudonana* PSI-FCPI structure (*Figure 1—figure supplement 4D, E*) are largely similar to those in the *C. gracilis* PSI-FCPI structure (*Nagao et al., 2020a*). However, Chl a102 of PsaI is found in the *T. pseudonana* PSI-FCPI structure but not in the *C. gracilis* PSI-FCPI structure (*Nagao et al., 2020a*), whereas a844 of PsaA and BCR843 of PsaB are identified in the *C. gracilis* PSI-FCPI structure (*Nagao et al., 2020a*) but not in the *T. pseudonana* PSI-FCPI structure. One of the Car molecules in PsaJ is identified as ZXT103 in the *T. pseudonana* PSI-FCPI structure, while it is BCR103 in the *C. gracilis* PSI-FCPI structure (*Nagao et al., 2020a*).

## Structure of the *T. pseudonana* FCPIs

Kumazawa et al. classified 44 *Lhc* genes in *T. pseudonana*, designating them as *Lhcf*, *Lhcq*, *Lhcr*, *Lhcx*, *Lhcz*, and *CgLhcr9* homologs (*Kumazawa et al., 2022*). Based on this classification, the five FCPI subunits in the PSI-FCPI structure are identified using five genes:

**Table 2.** Averaged *Q*-scores in each subunit.

| Subunit | Averaged *Q*-score | |
| --- | --- | --- |
| | Postprocessed map | Denoised map |
| PsaA | 0.84 | 0.83 |
| PsaB | 0.84 | 0.83 |
| PsaC | 0.87 | 0.85 |
| PsaD | 0.83 | 0.83 |
| PsaE | 0.80 | 0.80 |
| PsaF | 0.81 | 0.81 |
| PsaI | 0.83 | 0.82 |
| PsaJ | 0.81 | 0.81 |
| PsaL | 0.83 | 0.83 |
| PsaM | 0.83 | 0.82 |
| Psa29 | 0.65 | 0.70 |
| Unknown | 0.45 | 0.55 |
| FCPI-1 | 0.76 | 0.78 |
| FCPI-2 | 0.67 | 0.72 |
| FCPI-3 | 0.74 | 0.76 |
| FCPI-4 | 0.77 | 0.79 |
| FCPI-5 | 0.74 | 0.76 |

**Table 3.** Cofactors assigned in each subunit of the PSI-FCPI structure.

| Protein | Chlorophyll | Carotenoid | Lipid | Other |
|---|---|---|---|---|
| PsaA | 43 Chl *a*<br>1 Chl *a'* | 5 BCR | 2 LHG | 1 [4Fe-4S] cluster<br>1 phylloquinone |
| PsaB | 41 Chl *a* | 5 BCR | 1 LHG<br>1 DGD | 1 phylloquinone |
| PsaC | - | - | - | 2 [4Fe-4S] cluster |
| PsaD | - | - | - | - |
| PsaE | - | - | - | - |
| PsaF | 3 Chl *a* | 1 BCR | - | - |
| PsaI | 1 Chl *a* | 1 BCR | - | - |
| PsaJ | 1 Chl *a* | 1 BCR<br>1 ZXT | - | - |
| PsaL | 3 Chl *a* | 3 BCR | 1 LMG | - |
| PsaM | - | 1 BCR | 1 LHG | - |
| Psa29 | - | - | - | - |
| Unknown | 1 Chl *a* | 1 BCR | - | - |
| FCPI-1 | 7 Chl *a*<br>1 Chl *c* | 2 BCR<br>2 Fx<br>3 Ddx | 1 LHG | - |
| FCPI-2 | 10 Chl *a*<br>1 Chl *c* | 3 Fx<br>1 Ddx | - | - |
| FCPI-3 | 7 Chl *a*<br>3 Chl *c* | 2 Fx<br>2 Ddx | 1 LHG | - |
| FCPI-4 | 11 Chl *a*<br>2 Chl *c* | 4 Fx | 1 LHG | - |
| FCPI-5 | 10 Chl *a* | 4 Fx<br>1 Ddx | - | - |
| Total | 146 | 43 | 9 | 5 |

BCR = *β*-carotene. ZXT = zeaxanthin. Fx = fucoxanthin. Ddx = diadinoxanthin. Chl *a* = chlorophyll *a*. Chl *a'* = chlorophyll *a* epimer. Chl *c* = chlorophyll *c*. DGD = digalactosyl diacyl glycerol. LHG = dipalmitoyl phosphatidyl glycerol. LMG = distearoyl monogalactosyl diglyceride.

*RedCAP*, *Lhcr3*, *Lhcq10*, *Lhcf10*, and *Lhcq8*, corresponding to FCPI-1–5, respectively (*Figure 1A*). It is important to note that *RedCAP* is not included among the 44 *Lhc* genes (*Kumazawa et al., 2022*) but is classified within the LHC protein superfamily (*Engelken et al., 2010*; *Sturm et al., 2013*). For the assignment of each FCPI subunit, we focused on characteristic amino acid residues derived from their cryo-EM map, especially S61/V62/Q63 in FCPI-1; A70/R71/W72 in FCPI-2; Y64/R65/E66 in FCPI-3; M63/R64/Y65 in FCPI-4; and A62/R63/R64 in FCPI-5 (*Figure 1—figure supplement 5*). The root mean square deviations of the structures between FCPI-4 and the other four FCPIs range from 1.91 to 3.73 Å (*Table 4*).

Each FCPI subunit binds several Chl and Car molecules: 7 Chls *a*/1 Chl *c*/2 Fxs/3 Ddxs/2 BCRs in FCPI-1; 10 Chls *a*/1 Chl *c*/3 Fxs/1 Ddx in FCPI-2; 7 Chls *a*/3 Chls *c*/2 Fxs/2 Ddxs in FCPI-3; 11 Chls *a*/2 Chls *c*/4 Fxs in FCPI-4; and 10 Chls *a*/4 Fxs/1 Ddx in FCPI-5 (*Figure 1—figure supplement 6A–E* and *Table 3*). The axial ligands of the central Mg atoms of Chls within each FCPI are primarily provided by the main and side chains of amino acid residues (*Table 5*). Potential excitation-energy-transfer pathways can be proposed based on the close physical interactions among Chls between FCPI-3 and PsaA, between FCPI-3 and PsaL, between FCPI-1 and PsaI, and between FCPI-2 and PsaB (*Figure 1—figure supplement 7*).



**Figure 1.** Overall structure of the PSI-FCPI supercomplex from *T. pseudonana*. Structures are viewed from the stromal side (left panels) and from the direction perpendicular to the membrane normal (right panels). Only protein structures are depicted, with cofactors omitted for clarity. The FCPI (**A**) and PSI core (**B**) subunits are labeled and colored distinctly. The five FCPI subunits are labeled as FCPI-1–5 (red), with their corresponding gene products indicated in parentheses (black) in panel (**A**).

The online version of this article includes the following source data and figure supplement(s) for figure 1:

**Figure supplement 1.** Isolation and characterization of the *T. pseudonana* PSI-FCPI supercomplex.

**Figure supplement 1—source data 1.** PDF file containing the original SDS-PAGE gel for *Figure 1—figure supplement 1B*, with relevant bands indicated.

**Figure supplement 1—source data 2.** Original file of the SDS-PAGE gel shown in *Figure 1—figure supplement 1B*.

**Figure supplement 2.** Cryo-electron microscopy (cryo-EM) data collection and processing of PSI-FCPI.

**Figure supplement 3.** Evaluation of the cryo-electron microscopy (cryo-EM) map quality.

**Figure supplement 4.** Characteristic structures of the PSI subunits and pigments in the PSI-FCPI structure.

**Figure supplement 5.** Characteristic amino acid residues used for the identification of each FCPI subunit.

**Figure supplement 6.** Structures of FCPIs.

**Figure supplement 7.** Arrangement of pigment molecules within PSI-FCPI and possible excitation-energy-transfer pathways from FCPIs to PSI.

**Figure 2.** Structure and diversity of Psa29. (**A**) Structure of Psa29 depicted as cartoons. Psa29 was modeled from V47 to L178. (**B**) Cryo-electron microscopy (Cryo-EM) map of Psa29 and its surrounding environment, viewed from the stromal side. The red-squared areas are enlarged in panels (**C**) - (**G**). Yellow, PsaB; cyan, PsaC; blue, PsaD; magenta, PsaL; orange, Psa29. Protein–protein interactions of Psa29 with PsaB/PsaC (**C**), PsaC/PsaD (**D**), PsaB/PsaD

*Figure 2 continued on next page*

*Figure 2 continued*

(**E**), PsaB/PsaD/PsaL (**F**), and PsaD (**G**). Interactions are indicated by dashed lines, and the numbers are distances in Å. Amino acid residues participating in the interactions are labeled; for example, A71/C indicates Ala71 of PsaC. B, PsaB; C, PsaC; D, PsaD, L, PsaL; 29, Psa29. (**H**) Phylogenetic analysis of Psa29 in photosynthetic organisms. A maximum-likelihood tree of Psa29 proteins was inferred using IQ-TREE v2.2.2.7 with the WAG+F+G4 model and a trimmed alignment of 22 sequences comprising 245 amino acid residues. Numbers at the nodes represent ultrafast bootstrap support (%) (1000 replicates). The tree was mid-point rooted between diatoms and Bolidophyceae Parmales. Psa29 of *T. pseudonana* CCMP1335 is indicated by a red underline.

## Structural characteristics of RedCAP and its evolutionary implications

Among the FCPI subunits, only FCPI-1 contains two BCRs in addition to Fxs and Ddxs (*Figure 1—figure supplement 6A, F*). This is the first report of BCR binding to FCPIs in diatoms. FCPI-1 is identified as RedCAP, a member of the LHC protein superfamily but distinct from the LHC protein family (*Engelken et al., 2010*; *Sturm et al., 2013*); however, the functional and physiological roles of RedCAP remain unknown. FCPI-1 is positioned near PsaB, PsaI, and PsaL through protein–protein interactions with these subunits at both the stromal and lumenal sides (*Figure 3A*). At the stromal side, I138 and S139 of FCPI-1 interact with K121, G122, and F125 of PsaL (*Figure 3B*), whereas at the lumenal side, multiple interactions occur between I109 of FCPI-1 and F5 of PsaI, between T105/L106/T108 of FCPI-1 and W92/P94/F96 of PsaB, and between E102/W103 of FCPI-1 and S71/I73 of PsaL (*Figure 3C*). The protein–protein interactions at the lumenal side (*Figure 3C*) appear to be caused by a loop structure of FCPI-1 from Q96 to T116 (pink in *Figure 3D*), which is unique to FCPI-1 but absent in the other four FCPI subunits (pink in *Figure 3E*). This loop structure is inserted into a cavity formed by PsaB, PsaI, and PsaL (*Figure 3C, D*). These findings indicate that the Q96–T116 loop of FCPI-1 specifically recognizes and binds to the cavity provided by the PSI subunits.

RedCAP of *C. gracilis* (CgRedCAP) was not identified in the *C. gracilis* PSI-FCPI structures (*Nagao et al., 2020a*; *Xu et al., 2020*). As previously discussed (*Kato et al., 2024*), we proposed that CgRedCAP may bind to the *C. gracilis* PSI core at a site similar to LHCI-1 in the red alga *C. caldarium* PSI-LHCI through sequence analysis. This site corresponds to the FCPI-1 site in the PSI-FCPI of *T. pseudonana* in this study. A sequence alignment between RedCAP of *T. pseudonana* (TpRedCAP) and CgRedCAP is shown in *Figure 3—figure supplement 1A*, exhibiting a 72% sequence similarity. CgRedCAP contains a protein motif, Q106–I113 (QWGTLATI), corresponding to E102–I109 (EWGTLATI) in TpRedCAP (*Figure 3C*). These findings suggest the potential binding of CgRedCAP to PSI in *C. gracilis* at a position similar to FCPI-1 in the *T. pseudonana* PSI-FCPI structure. However, it remains unclear (1) whether CgRedCAP is indeed bound to the *C. gracilis* PSI-FCPI supercomplex and (2) if a loop structure corresponding to the Q96–T116 loop of TpRedCAP exists in CgRedCAP. Further structural studies of the *C. gracilis* PSI-FCPI are required to elucidate the molecular assembly mechanism of diatom RedCAPs.

RedCAPs have been found in the structures of PSI-LHCI in the red alga *Porphyridium purpureum* (*You et al., 2023*) and a PSI supercomplex with alloxanthin Chl *a/c*-binding proteins (PSI-ACPI) in the cryptophyte *Chroomonas placoidea* (*Zhao et al., 2023*), as summarized in our previous study (*Kato et al., 2024*). Both *P. purpureum* RedCAP (PpRedCAP) and *C. placoidea* RedCAP (CpRedCAP) exhibit loop structures similar to the Q96–T116 loop in TpRedCAP observed in the present study (*Figure 3—figure supplement 1B*). Multiple sequence alignments of TpRedCAP with PpRedCAP and CpRedCAP are shown in *Figure 3—figure supplement 1C*, revealing sequence similarities of 39% and 60%, respectively. PpRedCAP contains a protein motif of V105–L112 (VWGPLAQL), while CpRedCAP has a protein motif of Q117–A124 (QWGPLASA). These motifs correspond to E102–I109 (EWGTLATI) in TpRedCAP; however, the sequence conservation between TpRedCAP and PpRedCAP/CpRedCAP is lower than between TpRedCAP and CgRedCAP.

**Table 4.** FCPI proteins identified in the PSI-FCPI structure, their corresponding genes, and their root mean square deviation (RMSD) values compared with the FCPI-4 structure.

| Protein | Gene | RMSD (Å)/aligned Cα atoms |
|---|---|---|
| FCPI-1 | *RedCAP* | 3.73/95 |
| FCPI-2 | *Lhcr3* | 2.01/139 |
| FCPI-3 | *Lhcq10* | 2.02/139 |
| FCPI-4 | *Lhcf10* | 0.00/167 |
| FCPI-5 | *Lhcq8* | 1.91/128 |

**Table 5.** Chls and their ligands in each of the FCPI subunits.

| Protein | Chlorophyll/ligand |
| --- | --- |
| FCPI-1 | a301/E65, a302/N68, c304/H128, a305/H71, a306/E183, a308/N186, a311/w982*, a317/W215 |
| FCPI-2 | a301/E74, a302/H77, a303/Q91, a304/Q121, a305/E130, a306/E168, c307/-†, a308/H171, a309/Q185, a318/S35, a319/H184 |
| FCPI-3 | a301/E68, a302/N71, a303/w977*, c304/Q115, a305/E124, a306/E162, c307/w976*, a308/N165, a309/w978*, c310/D188 |
| FCPI-4 | a301/E67, c302/H70, a303/-†, a305/E122, a306/E162, a307/LHG330, c308/N165, a309/H179, a311/H100, a312/P91, a313/w980*, a314/Y196, a315/P142 |
| FCPI-5 | a301/E66, a302/N69, a303/w994*, a304/Q113, a305/E122, a306/E163, a307/E43, a308/N166, a309/S186, a316/E43 |

*Water molecules.
†The ligands of Chls may be water or lipid molecules which cannot be identified due to weak densities.

Among the four RedCAPs, the amino acids Trp, Gly, Leu, and Ala are conserved in the protein motifs (xWGxLAxx), implying that this conserved loop structure contributes to the binding of RedCAP to PSI across the red-lineage algae.

## Protein–protein interactions of the other FCPI subunits

FCPI-2 (Lhcr3) is positioned near PsaB and PsaM, engaging in protein–protein interactions with these subunits at distances of 3.0–4.3 Å at both the stromal and lumenal sides (***Figure 4***). The amino acid residues I63/T65/D66/Y69/W134/Y138/D140 of FCPI-2 are associated with W153/L154/K159/F160/W166 of PsaB at the stromal side (***Figure 4B***), while F116 and F120 of FCPI-2 interact with F5/I9/M12 of PsaM at the lumenal side (***Figure 4C***). The amino acid sequences corresponding to I63–Y69, F116–F120, and W134–D140 in Lhcr3 are not conserved in the Lhcr subfamily, comprising Lhcr1, Lhcr4, Lhcr7, Lhcr11, Lhcr12, Lhcr14, Lhcr17, Lhcr18, Lhcr19, and Lhcr20, as reported by ***Kumazawa et al., 2022*** (***Figure 4—figure supplement 1***).

FCPI-3 (Lhcq10) is positioned near PsaL, with protein–protein interactions at distances of 2.3–4.2 Å at the stromal side (***Figure 5A, B***). The amino acid residues L126/I130/L142/Y146/W147/V148/W155 of FCPI-3 are associated with F4/K6/P20/S25/L26/L30 of PsaL (***Figure 5B***). Given the homology between TpLhcq10 and CgLhcr9 (***Kumazawa et al., 2022***), we compared the amino acid sequence of Lhcq10 with the Lhcq and Lhcr subfamilies in *T. pseudonana* (***Figure 5—figure supplement 1A, B***). The sequence L126–W155 of Lhcq10 is not conserved in the Lhcq subfamily, comprising Lhcq1, Lhcq2, Lhcq3, Lhcq4, Lhcq5, Lhcq6, Lhcq7, Lhcq8, and Lhcq9 (***Figure 5—figure supplement 1A***), nor in the Lhcr subfamily, comprising Lhcr1, Lhcr3, Lhcr4, Lhcr7, Lhcr11, Lhcr12, Lhcr14, Lhcr17, Lhcr18, Lhcr19, and Lhcr20, as reported by ***Kumazawa et al., 2022*** (***Figure 5—figure supplement 1B***).

FCPI-4 (Lhcf10) is positioned near FCPI-5 through protein–protein interactions with it at distances of 2.6–3.6 Å at the lumenal side (***Figure 5A, C***). The amino acid residues Y196/P198/F199 of FCPI-4 interact with F82/F86/G87 of FCPI-5 (***Figure 5C***). The amino acid sequence Y196–F199 of Lhcf10 is not conserved in the Lhcf subfamily, comprising Lhcf1, Lhcf2, Lhcf3, Lhcf4, Lhcf5, Lhcf6, Lhcf7, Lhcf8, Lhcf9, Lhcf11, and Lhcf12, as reported by ***Kumazawa et al., 2022*** (***Figure 5—figure supplement 2***).

FCPI-5 (Lhcq8) is positioned near PsaL and FCPI-4 through protein–protein interactions at distances of 2.6–4.1 Å at both the stromal and lumenal sides (***Figure 5A, C, D***). The amino acid residues P108/Q109/A112/I115 of FCPI-5 interact with F134/I137/S141 of PsaL at the lumenal side (***Figure 5D***). The interactions between FCPI-5 and FCPI-4 are shown in ***Figure 5C***. The amino acid sequences F82–G87 and P107–I115 of Lhcq8 are not conserved in the Lhcq subfamily, comprising Lhcq1, Lhcq2, Lhcq3, Lhcq4, Lhcq5, Lhcq6, Lhcq7, Lhcq9, and Lhcq10, as reported by ***Kumazawa et al., 2022*** (***Figure 5—figure supplement 3A, B***).



**Figure 3.** Structural characteristics of FCPI-1 (RedCAP). (**A**) Interactions of FCPI-1 with PsaB, PsaI, and PsaL viewed from the stromal (left) and lumenal (right) sides. The areas encircled by black squares are enlarged in panels (**B**) and (**C**). Yellow, PsaB; magenta, PsaI; dark red, PsaL; red, FCPI-1. Protein–protein interactions of FCPI-1 with PsaL (**B**) and with PsaB/PsaI/PsaL (**C**). Interactions are indicated by dashed lines, and the numbers are distances in Å. Amino acid residues involved in the interactions are labeled; for example, S139/1 indicates Ser139 of FCPI-1. B, PsaB; I, PsaI; L, PsaL; 1, FCPI-1. (**D**) Characteristic loop structure from Q96 to T116 in FCPI-1, viewed from the lumenal side. Q96 and T116 are labeled with sticks, and the Q96–T116 loop is colored pink. (**E**) Superpositions of FCPI-1 with FCPI-2, FCPI-3, FCPI-4, and FCPI-5. Only proteins are depicted. Q96 and T116 in the Q96–T116 loop of FCPI-1 are shown with sticks.

The online version of this article includes the following figure supplement(s) for figure 3:

**Figure supplement 1.** Characteristics of the sequences and structures of RedCAPs in the red-lineage algae.

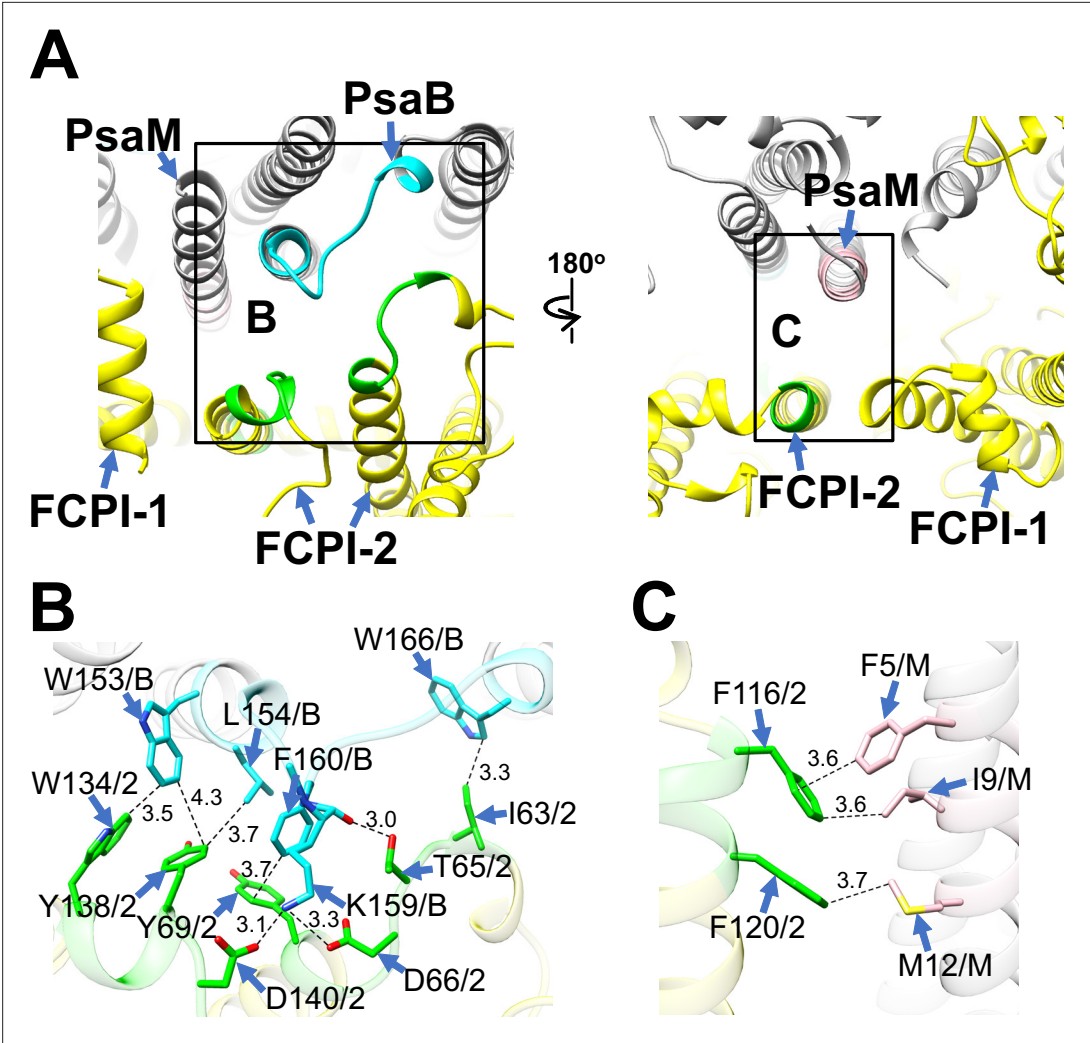

**Figure 4.** Structural characteristics of FCPI-2. (**A**) Interactions of FCPI-2 with PsaB and PsaM viewed from the stromal (left) and lumenal (right) sides. The areas encircled by black squares are enlarged in panels (**B**) and (**C**). PSI subunits are colored gray, and FCPI subunits are colored yellow. Protein–protein interactions are shown in different colors: green, FCPI-2; cyan, PsaB; pink, PsaM. Protein–protein interactions of FCPI-2 with PsaB (**B**) and PsaM (**C**). Interactions are indicated by dashed lines, and the numbers represent distances in Å. Amino acid residues involved in the interactions are labeled; for example, Y138/2 indicates Tyr138 of FCPI-2. B, PsaB; M, PsaM; 2, FCPI-2.

The online version of this article includes the following figure supplement(s) for figure 4:

**Figure supplement 1.** Comparisons of the sequence of Lhcr3 (FCPI-2) with those of the Lhcr subfamily in *T. pseudonana*.

## Molecular insights into the assembly of FCPIs in diatom PSI-FCPI supercomplexes

To evaluate the molecular assembly of FCPI subunits in the *T. pseudonana* PSI-FCPI structure, we focused on protein–protein interactions based on their close proximities (*Figures 3–5*) and the amino acid residues in non-conserved regions among 44 FCPs (*Figure 4—figure supplement 1*, *Figure 5— figure supplements 1–3*). This approach is based on the premise that selective associations of FCPIs with PSI require specific amino acid residues unique to each FCPI. Protein–protein interactions among FCPI subunits, as well as between FCPI and PSI subunits, occur at both the stromal and lumenal sides (*Figures 3–5*), and are likely recognized by unique amino acid residues of FCPIs that are not conserved in each LHC subfamily (*Figure 4—figure supplement 1*, *Figure 5—figure supplements 1–3*). Thus, the binding and assembly of each FCPI subunit to PSI are likely determined by the amino acid sequences within the loop regions of the 44 FCPs in *T. pseudonana*.



**Figure 5.** Structural characteristics of FCPI-3, 4, and 5. (**A**) Interactions among FCPIs and between FCPIs and PsaL, viewed from the stromal (left) and lumenal (right) sides. The areas encircled by black squares are enlarged in panels (**B**)-(**D**). Photosystem I (PSI) subunits are colored gray, and FCPI subunits are colored yellow. Protein–protein interactions are shown in different colors: blue, FCPI-3; magenta, FCPI-4; orange, FCPI-5; purple, PsaL. Protein–protein interactions between FCPI-3 and PsaL (**B**), between FCPI-4 and FCPI-5 (**C**), and between FCPI-5 and PsaL (**D**). Interactions are indicated by dashed lines, and the numbers represent distances in Å. Amino acid residues involved in the interactions are labeled; for example, L126/3 indicates Leu126 of FCPI-3. L, PsaL; 3, FCPI-3; 4, FCPI-4; 5, FCPI-5.

The online version of this article includes the following figure supplement(s) for figure 5:

*Figure 5 continued on next page*

*Figure 5 continued*

**Figure supplement 1.** Comparisons of the sequence of Lhcq10 (FCPI-3) with those of the Lhcq and Lhcr subfamilies in *T.pseudonana*.

**Figure supplement 2.** Comparisons of the sequence of Lhcf10 (FCPI-4) with those of the Lhcf subfamily in *T. pseudonana*.

**Figure supplement 3.** Comparisons of the sequence of Lhcq8 (FCPI-5) with those of the Lhcq subfamily in *T. pseudonana*.

The diatom *C. gracilis* exhibits two distinct PSI-FCPI structures: one with 16 FCPI subunits (*Nagao et al., 2020a*) and the other with 24 FCPI subunits (*Xu et al., 2020*). These structural variations arise from changes in the antenna sizes of FCPIs within the *C. gracilis* PSI-FCPI supercomplexes, in response to varying growth conditions, especially $CO_2$ concentrations and temperatures (*Nagao et al., 2020b*). Notably, the *C. gracilis* PSI-FCPI structure contains five FCPI subunits located at the same binding sites as FCPI-1–5 in the *T. pseudonana* PSI-FCPI structure (*Figure 6A*). A summary of the relationship between the *Lhc* genes encoding FCPs, the distinct gene *RedCAP*, and the binding positions of FCPI-1–5 in *T. pseudonana* and *C. gracilis* is shown in *Figure 6B*. The gene nomenclature for the *C. gracilis* FCPIs follows the conventions established by *Kumazawa et al., 2022*, as discussed in our recent study (*Kato et al., 2024*).

Phylogenetic analysis clearly showed that at the FCPI-1, 2, 3, and 5 sites in the *T. pseudonana* PSI-FCPI structure, TpRedCAP, TpLhcr3, TpLhcq10, and TpLhcq8 are orthologous to CgRedCAP, CgLhcr1, CgLhcr9, and CgLhcq12, respectively (*Figure 6C*). The characteristic protein loops of TpRedCAP and CpRedCAP likely participate in interactions with PSI at the FCPI-1 site, as noted above (*Figure 3—figure supplement 1*). At the FCPI-2 site, comparative analyses revealed that the amino acid residues facilitating interactions between TpLhcr3 and TpPsaB/TpPsaM closely parallel those observed in the CgLhcr1-CgPsaB and CgLhcr1-CgPsaM pairs (*Figure 6—figure supplement 1*). Similarly, a high degree of similarity characterized the residues involved in the interaction pairs of TpLhcq10-TpPsaL/CgLhcr9-CgPsaL at the FCPI-3 site (*Figure 6—figure supplement 2A, B*), as well as TpLhcq8-TpPsaL/CgLhcq12-CgPsaL at the FCPI-5 site (*Figure 6—figure supplement 2C, D*), underscoring the conserved nature of these interactions. However, TpLhcf10 is not homologous to CgLhcf3 (*Figure 6C*), despite both being located at the FCPI-4 site in their respective PSI-FCPI structures (*Figure 6A*). These findings suggest that the two diatoms possess both a conserved mechanism of protein–protein interactions across characteristic protein motifs between FCPI and PSI subunits, and a different mechanism of interactions among FCPIs.

It is notable that the *C. gracilis* PSI-FCPI structure binds remarkably more FCPI subunits than that of *T. pseudonana*, for example, 16 or 24 subunits in *C. gracilis* as reported in the previous studies (*Nagao et al., 2020a*; *Xu et al., 2020*), versus 5 subunits in *T. pseudonana* in the present study. The reason for this difference remains unclear. One possibility is that some FCPI subunits are released during detergent solubilization in *T. pseudonana*, while they are retained in *C. gracilis*. Alternatively, the number of FCPI subunits may be inherently lower in *T. pseudonana*, which may reflect adaptations to different living environments. Further studies are needed to resolve this question.

## Extension to molecular assembly of PSI-LHCI supercomplexes

The mechanisms of protein–protein interactions in diatom PSI-FCPI supercomplexes are likely developed by the specific binding of FCPs selected from 44 TpFCPs and 46 CgFCPs in addition to RedCAPs. Like a lock-and-key mechanism, one FCP cannot be substituted by another in forming the PSI-FCPI supercomplexes in the two diatoms; for example, TpLhcq10 binds specifically at the FCPI-3 site but not at the other sites such as FCPI-2. This selective binding mechanism of FCPIs may dictate the molecular assembly of PSI-FCPI. Importantly, the selective binding of FCPIs was identified for the first time by comparing the structures of PSI-FCPI supercomplexes and the amino acid sequences of FCPIs between the two diatom species. This approach can be extended to the LHC protein superfamily in the green and red lineages, enabling comparisons of protein structures and sequences of PSI-LHCI supercomplexes among closely related species. This, in turn, lays the foundation for elucidating the underlying mechanism of PSI-LHCI supercomplex assembly. Thus, this study will shed light on answering the evolutionary question of how LHCIs recognize their binding sites at PSI in photosynthetic organisms.

**Figure 6.** Comparisons of structures and sequences of FCPIs in the PSI-FCPI structures between *T. pseudonana* and *C. gracilis*. (**A**) Superposition of the PSI-FCPI structures between *T. pseudonana* and *C. gracilis* (PDB: 6LY5). FCPI subunits from *T. pseudonana* and *C. gracilis* are colored red and cyan, respectively. The structures are viewed from the stromal side. The FCPI-1–5 sites are labeled. (**B**) Correlation of the names of FCPIs in the structures with their corresponding genes between *T. pseudonana* and *C. gracilis*. The FCPI genes are derived from ***Kumazawa et al., 2022*** and ***Kato et al., 2024*** for *C. gracilis*. (**C**) Phylogenetic analysis of FCPs and RedCAPs from *T. pseudonana* (Tp) and *C. gracilis* (Cg). In addition to the RedCAP family, 44 TpFCPs and 46 CgFCPs are grouped into five Lhc subfamilies and CgLhcr9 homologs. Maroon, RedCAP family; magenta, Lhcq subfamily; red, Lhcz subfamily; orange, Lhcr subfamily; brown, CgLhcr9 homologs; green, Lhcf subfamily; blue, Lhcx subfamily. The FCPs and RedCAPs located at the FCPI-1–5 sites

*Figure 6 continued on next page*

*Figure 6 continued*

are labeled. The tree was inferred using IQ-TREE 2 (*Minh et al., 2020*) with the Q.pfam + R4 model selected by ModelFinder (*Kalyaanamoorthy et al., 2017*). The light purple circular symbols on the tree represent bootstrap support (%).

The online version of this article includes the following figure supplement(s) for figure 6:

**Figure supplement 1.** Comparisons of amino acid sequences between FCPs and photosystem I (PSI) proteins at the FCPI-2 site in the *T. pseudonana* PSI-FCPI structure.

**Figure supplement 2.** Comparisons of amino acid sequences between FCPs and PsaL proteins at the FCPI-3 and FCPI-5 sites in the *T. pseudonana* PSI-FCPI structure.

## Methods

### Cell growth and preparation of thylakoid membranes

The marine centric diatom *T. pseudonana* CCMP1335 was grown in artificial seawater supplemented with sodium metasilicate and KW21 (*Nagao et al., 2007*) at 20°C under a photosynthetic photon flux density of 30 μmol photons $m^{-2} s^{-1}$ provided by white LED, with bubbling of air containing 3% (vol/vol) $CO_2$. The cells were harvested by centrifugation, disrupted by agitation with glass beads (*Nagao et al., 2017*), and the thylakoid membranes were pelleted by further centrifugation. The resulting thylakoid membranes were suspended in 50 mM Mes-NaOH (pH 6.5) buffer containing 1 M betaine and 1 mM ethylenediaminetetraacetic acid (EDTA).

### Purification of the PSI-FCPI supercomplex

Thylakoid membranes were solubilized with 1% (wt/vol) *n*-dodecyl-*β*-D-maltoside (*β*-DDM) at a Chl concentration of 0.5 mg $ml^{-1}$ for 20 min on ice in the dark with gentle stirring. After centrifugation at 162,000 × *g* for 20 min at 4°C, the supernatant was loaded onto a Q-Sepharose anion-exchange column (1.6 cm inner diameter, 25 cm length) equilibrated with 20 mM Mes-NaOH (pH 6.5) buffer containing 0.2 M trehalose, 5 mM $CaCl_2$, 10 mM $MgCl_2$, and 0.03% *β*-DDM (buffer A). The column was washed with buffer A until the eluate became colorless. Elution was performed at a flow rate of 1.0 ml $min^{-1}$ using a linear gradient of buffer A and buffer B (buffer A plus 500 mM NaCl) with the following time and gradient: 0–600 min, 0–60% buffer B; 600–800 min, 60–100% buffer B; 800–900 min, 100% buffer B. The PSI-FCPI-enriched fraction was eluted at 194–247 mM NaCl, then collected and subsequently loaded onto a linear gradient containing 10–40% (wt/vol) trehalose in 20 mM Mes-NaOH (pH 6.5) buffer containing 5 mM $CaCl_2$, 10 mM $MgCl_2$, 100 mM NaCl, and 0.03% *β*-DDM. After centrifugation at 154,000 × *g* for 18 hr at 4°C (P40ST rotor; Hitachi), a green fraction (*Figure 1—figure supplement 1A*) was collected and concentrated using a 150-kDa cut-off filter (Apollo; Orbital Biosciences) at 4000 × *g*. The concentrated samples were stored in liquid nitrogen until use.

### Biochemical and spectroscopic analyses of the PSI-FCPI supercomplex

The polypeptide bands of PSI-FCPI were analyzed by sodium dodecyl sulfate–polyacrylamide gel electrophoresis with 16% (wt/vol) acrylamide and 7.5 M urea, following the method of *Ikeuchi and Inoue, 1988* (*Figure 1—figure supplement 1B*, *Figure 1—figure supplement 1—source data 1 and 2*). The PSI-FCPI supercomplexes (4 μg of Chl) were solubilized in 3% lithium lauryl sulfate and 75 mM dithiothreitol for 10 min at 60°C, and then loaded onto the gel. A standard molecular weight marker (SP-0110; APRO Science) was used. The absorption spectrum of PSI-FCPI was measured at room temperature using a UV-Vis spectrophotometer (UV-2450; Shimadzu) (*Figure 1—figure supplement 1C*), and the fluorescence emission spectrum of PSI-FCPI was measured at 77 K upon excitation at 430 nm using a spectrofluorometer (RF-5300PC; Shimadzu) (*Figure 1—figure supplement 1D*). The pigment composition of PSI-FCPI was analyzed by high-performance liquid chromatography following the method of *Nagao et al., 2013*, and the elution profile was monitored at 440 nm (*Figure 1—figure supplement 1E*).

### Cryo-EM data collection

A 3 μl aliquot of the *T. pseudonana* PSI-FCPI supercomplex (3.0 mg of Chl $ml^{-1}$) in 20 mM Mes-NaOH (pH 6.5) buffer containing 0.5 M betaine, 5 mM $CaCl_2$, 10 mM $MgCl_2$, and 0.03% *β*-DDM was applied to Quantifoil R1.2/1.3 Cu 300 mesh grids in the chamber of FEI Vitrobot Mark IV (Thermo Fisher Scientific). The grid was then blotted with filter paper for 4 s at 4°C under 100% humidity and plunged

into liquid ethane cooled by liquid nitrogen. The frozen grid was transferred to a CRYO ARM 300 electron microscope (JEOL) equipped with a cold-field emission gun operated at 300 kV. All image stacks were collected from 5 × 5 holes per stage adjustment to the central hole and image shifts were applied to the surrounding holes while maintaining an axial coma-free condition. The images were recorded using an in-column energy filter with a slit width of 20 eV at a nominal magnification of ×60,000 on a direct electron detector (Gatan K3, AMETEK). The nominal defocus range was −1.8 to −1.2 μm, and the physical pixel size corresponded to 0.752 Å. Each image stack was exposed at a dose rate of 21.46 e⁻ Å⁻² s⁻¹ for 2.33 s in CDS mode, with dose-fractionated 50 movie frames. A total of 8,950 image stacks were collected.

**Table 6.** Correspondence of the numbering of pigments in each PSI core subunit described in the text with those in the PDB file.

|  | PsaI | PsaJ | PsaL |
|---|---|---|---|
| Chls in the text | PDB No. (Chain ID) | PDB No. (Chain ID) | PDB No. (Chain ID) |
| 102 | 301 (1)* |  |  |
| 203 |  |  | 204 (L) |
| Car in the text |  |  |  |
| 103 |  | 105 (J) |  |

*Chain in the adjacent unit.

## Cryo-EM image processing

The resultant movie frames were aligned and summed using MotionCor2 (*Zheng et al., 2017*) to produce dose-weighted images. The contrast transfer function (CTF) estimation was performed using CTFFIND4 (*Mindell and Grigorieff, 2003*). All subsequent processes were carried out using RELION-4.0 (*Kimanius et al., 2021*). A total of 2,733,572 particles were automatically picked and subjected to reference-free 2D classification. From these, 1,132,721 particles were selected from well-defined 2D classes and further processed for 3D classification without imposing any symmetry. An initial model for the first 3D classification was generated *de novo* from the 2D classification. A 240-Å spherical mask was used during the 3D classification and refinement processes. As illustrated in *Figure 1—figure supplement 2C*, the final PSI-FCPI structure was reconstructed from 75,667 particles. The overall resolution of the cryo-EM map was determined to be 2.30 Å, based on the gold-standard FSC curve with a cut-off value of 0.143 (*Figure 1—figure supplement 3A*; *Grigorieff and Harrison, 2011*). Local resolutions were calculated using RELION (*Figure 1—figure supplement 3C*).

## Model building and refinement

Two types of the cryo-EM maps were employed for the model building of the PSI-FCPI supercomplex: a postprocessed map and a denoised map generated using Topaz version 0.2.4 (*Bepler et al., 2020*). The postprocessed map was denoised using a trained model over 100 epochs using two half-maps. Initial models of each subunit in the PSI-FCPI supercomplex were generated by ModelAngelo (*Jamali et al., 2024*) and subsequently inspected and manually adjusted against the maps with Coot (*Emsley et al., 2010*). Each model was built based on interpretable features from the density maps at a contour level of 2.5 σ in both the denoised and postprocessed maps. For the assignment of Chls, Chls *a* and *c* were distinguished by inspecting the density map corresponding to the phytol chain at the least level not to link the map of Chls with that of noise. All Chls *c* were assigned as Chl *c*1 due to the inability to distinguish between Chl *c*1 and Chl *c*2 at the present resolution. For the assignment of Cars, Fx and Ddx were distinguished based on the density surrounding the head groups of Cars with the above threshold. The PSI-FCPI structure was refined using phenix.real_space_refine (*Adams et al., 2010*) and Servalcat (*Yamashita et al., 2021*), incorporating geometric restraints for protein-cofactor coordination. The final model was validated with MolProbity (*Chen et al., 2010*), EMRinger (*Barad et al., 2015*), and *Q*-score (*Pintilie et al., 2020*). The statistics for all data collection and structure refinement are summarized in *Tables 1 and 2*. All structural figures were prepared using PyMOL (*Schrödinger, 2021*), UCSF Chimera (*Pettersen et al., 2004*), and UCSF ChimeraX (*Pettersen et al., 2021*). Since the numbering of Chls, Cars, and other cofactors in this paper differs from those in the PDB data, the corresponding relationships are provided in *Tables 6–8*.

**Table 7.** Correspondence of the numbering of pigments in each FCPI subunit described in the text with those in the PDB file.

| | FCPI-1 | FCPI-2 | FCPI-3 | FCPI-4 | FCPI-5 |
|---|---|---|---|---|---|
| Chls in the text | PDB No. (Chain ID) | PDB No. (Chain ID) | PDB No. (Chain ID) | PDB No. (Chain ID) | PDB No. (Chain ID) |
| 301 | 303 (1) | 205 (2) | 202 (3) | | 207 (5) |
| 302 | 304 (1) | 206 (2) | 203 (3) | | 208 (5) |
| 303 | | 207 (2) | 204 (3) | | 209 (5) |
| 304 | 305 (1) | 208 (2) | 205 (3) | | 210 (5) |
| 305 | 306 (1) | 209 (2) | 206 (3) | 304 (4) | 211 (5) |
| 306 | 307 (1) | 210 (2) | 207 (3) | 305 (4) | 212 (5) |
| 307 | | 211 (2) | 208 (3) | 306 (4) | 213 (5) |
| 308 | | 212 (2) | 209 (3) | 307 (4) | 214 (5) |
| 309 | | 213 (2) | 210 (3) | 308 (4) | 215 (5) |
| 310 | | | 211 (3) | | |
| 311 | 309 (1) | | | 309 (4) | |
| 312 | | | | 310 (4) | |
| 313 | | | | 311 (4) | |
| 314 | | | | 312 (4) | |
| 315 | | | | 313 (4) | |
| 316 | | | | | 216 (5) |
| 317 | 310 (1) | | | | |
| 318 | | 214 (2) | | | |
| 319 | | 215 (2) | | | |
| | | | | | |
| Cars in the text | | | | | |
| 321 | 311 (1) | 216 (2) | 212 (3) | 314 (4) | 217 (5) |
| 322 | 312 (1) | 217 (2) | 213 (3) | 315 (4) | 218 (5) |
| 323 | | | 209 (L)* | | |
| 324 | 313 (1) | 218 (2) | 214 (3) | 316 (4) | 219 (5) |
| 325 | 314 (1) | 219 (2) | | 228 (3)* | 220 (5) |
| 326 | 315 (1) | | | | 221 (5) |
| 327 | 316 (1) | | | | |
| 328 | 317 (1) | | | | |

*Chain in the adjacent unit.

## Phylogenetic analysis

Amino acid sequences were aligned using MAFFT L-INS-i v7.490 or MAFFT E-INS-i v7.520 (*Katoh and Standley, 2013*). The alignment was trimmed using ClipKit v1.4.1 with the smart-gap mode. Phylogenetic trees were inferred using IQ-TREE 2 (*Minh et al., 2020*) with the model selected by ModelFinder (*Kalyaanamoorthy et al., 2017*). The trees were visualized using iTOL v6 (*Letunic and Bork, 2021*). Ultrafast bootstrap approximation was performed with 1000 replicates (*Hoang et al., 2018*).

**Table 8.** Correspondence of the numbering of other cofactors described in the text with those in the PDB file.

| Waters in the text | PDB No. (Chain ID) |
|---|---|
| 976 | 317 (3) |
| 977 | 313 (3) |
| 978 | 316 (3) |
| 980 | 402 (4) |
| 982 | 425 (1) |
| 994 | 308 (5) |
| | |
| Lipid in the text | |
| 330 | 317 (4) |

## Acknowledgements

We thank Kumiyo Kato and Satoko Kakiuchi for their assistance in this study. The cells of *T. pseudonana* CCMP1335 were given by Prof. Yusuke Matsuda, Kwansei Gakuin University, Japan. Cryo-EM data was obtained using EM01CT and EM02CT of SPring-8 with the approval of the Japan Synchrotron Radiation Research Institute (JASRI Proposal No. 2022B2728 (J-RS) and No. 2023A2715 (YN)). This work was supported by JSPS KAKENHI grant Nos. JP22KJ2017 (MK), JP23K14211 (YN), JP22H04916 (J-RS), JP23H02347 (KI), and JP23H02423 (RN), Takeda Science Foundation (RN, KK), and Research Support Project for Life Science and Drug Discovery (Basis for Supporting Innovative Drug Discovery and Life Science Research (BINDS)) from AMED under support No. 4176 (J-RS).

## Additional information

### Funding

| Funder | Grant reference number | Author |
|---|---|---|
| Japan Society for the Promotion of Science | JP22KJ2017 | Minoru Kumazawa |
| Japan Society for the Promotion of Science | JP23K14211 | Yoshiki Nakajima |
| Japan Society for the Promotion of Science | JP22H04916 | Jian-Ren Shen |
| Japan Society for the Promotion of Science | JP23H02347 | Kentaro Ifuku |
| Japan Society for the Promotion of Science | JP23H02423 | Ryo Nagao |
| Takeda Science Foundation | | Koji Kato Ryo Nagao |

The funders had no role in study design, data collection, and interpretation, or the decision to submit the work for publication.

### Author contributions

Koji Kato, Yoshiki Nakajima, Data curation, Formal analysis, Funding acquisition, Validation, Investigation, Writing – original draft; Jian Xing, Formal analysis, Validation, Investigation, Writing – original draft; Minoru Kumazawa, Formal analysis, Funding acquisition, Investigation; Haruya Ogawa, Formal analysis, Investigation; Jian-Ren Shen, Resources, Supervision, Funding acquisition, Writing – original draft; Kentaro Ifuku, Resources, Supervision, Funding acquisition, Validation, Writing – original draft; Ryo Nagao, Conceptualization, Resources, Data curation, Formal analysis, Supervision, Funding acquisition, Validation, Investigation, Writing – original draft, Project administration, Writing – review and editing

### Author ORCIDs
Jian-Ren Shen https://orcid.org/0000-0003-4471-8797
Kentaro Ifuku https://orcid.org/0000-0003-0241-8008
Ryo Nagao https://orcid.org/0000-0001-8212-3001

Reviewer #1 (Public review): https://doi.org/10.7554/eLife.99858.3.sa1
Author response https://doi.org/10.7554/eLife.99858.3.sa2

## Additional files

### Supplementary files
• MDAR checklist

### Data availability
Atomic coordinates, cryo-EM maps, and raw image data for the reported structure have been deposited in the Protein Data Bank under an accession code 8XLS (https://www.rcsb.org/structure/8XLS), in the Electron Microscopy Data Bank under an accession code EMD-38457 (https://www.ebi.ac.uk/emdb/EMD-38457), and in the Electron Microscopy Public Image Archive under an accession code EMPIAR-12142 (https://www.ebi.ac.uk/empiar/EMPIAR-12142/), respectively.

The following datasets were generated:

| Author(s) | Year | Dataset title | Dataset URL | Database and Identifier |
|---|---|---|---|---|
| Kato K, Nakajima Y, Shen J-R, Nagao R | 2024 | PSI-FCPI of the diatom *Thalassiosira pseudonana* CCMP1335 | https://www.rcsb.org/structure/8XLS | RCSB Protein Data Bank, 8XLS |
| Kato K, Nakajima Y, Shen J-R, Nagao R | 2024 | PSI-FCPI of the diatom *Thalassiosira pseudonana* CCMP1335 | https://www.ebi.ac.uk/emdb/EMD-38457 | Electron Microscopy Data Bank, EMD-38457 |
| Kato K, Nakajima Y, Shen J-R, Nagao R | 2024 | PSI-FCPI of the diatom *Thalassiosira pseudonana* CCMP1335 | https://www.ebi.ac.uk/empiar/EMPIAR-12142/ | Electron Microscopy Public Image Archive, EMPIAR-12142 |

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
