## [Editor Report · eLife Assessment]

This **important** study presents a high-resolution cryoEM structure of the supercomplex between photosystem I (PSI) and fucoxanthin chlorophyll *a*/*c*-binding proteins (FCPs) from the model diatom *Thalassiosira pseudonana* CCMP1335, revealing subunits, protein:protein interactions and pigments not previously seen in other diatoms or red/green photosynthetic lineages. Combining structural, sequence and phylogenetic analyses, the authors provide **convincing** evidence of conserved motifs crucial for the binding of FCPIs, accompanied by interesting speculation about the mechanisms governing the assembly of PSI-FCPI supercomplexes in diatoms and their implications for related PSI-LHCI supercomplexes in plants. The findings set the stage for functional experiments that will further advance the fields of photosynthesis, bioenergy, ocean biogeochemistry and evolutionary relationships between photosynthetic organisms.

---

## [Referee Report · Reviewer #1 (Public review)]

The authors present the cryo-EM structure of PSI-fucoxanthin chlorophyll a/c-binding proteins (FCPs) supercomplex from the diatom Thalassiosira pseudonana CCMP1335 at a global resolution of 2.3 Å. This exceptional resolution allows the authors to construct a near-atomic model of the entire supercomplex and elucidate the molecular details of FCPs arrangement. The high-resolution structure reveals subunits not previously identified in earlier reconstructions and models, as well as sequence analysis of PSI-FCPIs from other diatoms and red algae. Additionally, the authors use their model in conjunction with a phylogenetic analysis to compare and contrast the structural features of the T. pseudonana supercomplex with those of Chaetoceros gracilis, uncovering key structural features that contribute to the efficiency of light energy conversion in diatoms.

The study employs the advanced technique of single particle cryo-electron microscopy to visualize the complex architecture of the PSI supercomplex at near-atomic resolution and analyze the specific roles of FCPs in enhancing photosynthetic performance in diatoms.

Overall, the approach and data are both compelling and of high quality. The paper is well written and will be of wide interest for comprehending the molecular mechanisms of photosynthesis in diatoms. This work provides valuable insights for applications in bioenergy, environmental conservation, plant physiology, and membrane protein structural biology.

---

## [Author Response]

The following is the authors’ response to the original reviews.

**Public Reviews:**

**Reviewer #1 (Public Review):**
The authors present the cryo-EM structure of of PSI-fucoxanthin chlorophyll a/c-binding proteins (FCPs) supercomplex from the diatom Thalassiosira pseudonana CCMP1335 at a global resolution of 2.3 Å. This exceptional resolution allows the authors to construct a near-atomic model of the entire supercomplex and elucidate the molecular details of FCPs arrangement. The high-resolution structure reveals subunits not previously identified in earlier reconstructions and models, as well as sequence analysis of PSI-FCPIs from other diatoms and red algae. Additionally, the authors use their model in conjunction with a phylogenetic analysis to compare and contrast the structural features of the T. pseudonana supercomplex with those of Chaetoceros gracilis, uncovering key structural features that contribute to the efficiency of light energy conversion in diatoms.The study employs the advanced technique of single particle cryo-electron microscopy to visualize the complex architecture of the PSI supercomplex at near-atomic resolution and analyze the specific roles of FCPs in enhancing photosynthetic performance in diatoms.Overall, the approach and data are both compelling and of high quality. The paper is well written and will be of wide interest for comprehending the molecular mechanisms of photosynthesis in diatoms. This work provides valuable insights for applications in bioenergy, environmental conservation, plant physiology, and membrane protein structural biology.

We thank you very much for your highly positive evaluation and comments on our manuscript.

**Reviewer #2 (Public Review):**
Summary:This manuscript elucidated the cryo-electron microscopic structure of a PSI supercomplex incorporating fucoxanthin chlorophyll a/c-binding proteins (FCPs), designated as PSI-FCPI, isolated from the diatom Thalassiosira pseudonana CCMP1335. Combining structural, sequence, and phylogenetic analyses, the authors provided solid evidence to reveal the evolutionary conservation of protein motifs crucial for the selective binding of individual FCPI subunits and provided valuable information about the molecular mechanisms governing the assembly and selective binding of FCPIs in diatoms.Strengths:The manuscript is well-written and presented clearly as well as consistently. The supplemental figures are also of high quality.Weaknesses:Only minor comments (provided in recommendations for authors) to help improve the manuscript.

We thank you very much for your highly positive evaluation and comments on our manuscript.

**Reviewer #3 (Public Review):**
Summary:Understanding the structure and function of the photosynthetic machinery is crucial for grasping its mode of action. Photosystem I (PSI) plays a vital role in light-driven electron transfer, which is essential for generating cellular reducing power. A primary strategy to mitigate light and environmental stresses involves incorporating peripheral light-harvesting proteins. Among various lineages, the number of LHCIs and their protein and pigment compositions differ significantly in PSI-LHCI structures. However, it is still unclear how LHCIs recognize their specific binding sites in the PSI core. This study aims to address this question by obtaining a high-resolution structure of the PSI supercomplex, including fucoxanthin chlorophyll a/c-binding proteins (FCPs), referred to as PSI-FCPI, isolated from the diatom Thalassiosira pseudonana. Through structural and sequence analyses, distinct protein-protein interactions are identified at the interfaces between FCPI and PSI subunits, as well as among FCPI subunits themselves.Strengths:The primary strength of this work lies in its superb isolation and structural determination, followed by clear discussion and conclusions. However, the interactions among the protein complexes and their relevance in formulating general rules are not definitively established. While efficiency is a crucial aspect, preventing damage is equally important, and currently, we cannot infer this from the provided structures.Weaknesses:The interactions among the protein complexes and their relevance in formulating general rules are not definitively established. While efficiency is a crucial aspect, preventing damage is equally important, and currently, we cannot infer this from the provided structures.

We thank you very much for your highly positive evaluation and comments on our manuscript. This study is aimed to decipher the interactions among different protein subunits within the PSI-FCPI supercomplex, from which we wish to draw their relevance in formulating general rules. While we agree that damage is equally important, it is unclear to us what kind of damage you are mentioning, and we consider that this may need to be treated in another publication, as we cannot elucidate everything in one paper.

**Recommendations for the authors:**

**Reviewer #1 (Recommendations For The Authors):**
(1) Line 69: "Diatoms are one of the most important phytoplankton in aquatic environments and contribute to the primary production in the ocean remarkably." Check the sentence, something is missing.

We modified the sentence as follow:

"Diatoms are among the most essential phytoplankton in aquatic environments, playing a crucial role in the global carbon cycle, supporting marine food webs, and contributing significantly to nutrient cycling, thus ensuring the health and sustainability of marine ecosystems"

(2) Supplementary Figure 1B: The SDS-PAGE gel shows multiple bands. Do the authors know the identity of these proteins, or have they considered analyzing the bands using mass spectrometry? The band at ~17 kDa is particularly intense. Could you comment on this? Have you tried running a Native-PAGE gel?

We did not identify protein bands by MS analysis. The protein bands in the PSI-FCPI supercomplex of this diatom have been identified by Ikeda et al. 2013. The protein bands of our sample were similar to those of Ikeda et al. 2013. To explain this, we modified the sentences and cited Ikeda et al. 2013 in the revised manuscript (lines 89-91).

"The PSI-FCPI supercomplexes were purified from the diatom *T. pseudonana* CCMP1335 and analyzed by biochemical and spectroscopic techniques (Fig. S1). Notably, the protein bands of PSI-FCPI closely resembled those reported in a previous study (31)."

The ~17 kDa protein band appears to be FCPIs, which was identified in Ikeda et al. 2013. We did not perform BN-PAGE of this sample; however, we performed trehalose density gradient centrifugation (Fig. S1A).

(3) Can the authors comment on the position of the FCPI subunits in the PSI supercomplex in diatoms compared to the arrangement of LHCIs in complex with PSI in cyanobacteria, green algae, and angiosperms? This information would be useful to incorporate into the text.

We previously compared the PSI-FCPI structures of the diatom *C. gracilis* to the PSI-LHCI structures of land plant, green alga, and red alga (Nagao et al., 2020). Also, Xu et al. 2020 compared the *C. gracilis* PSI-FCPI structure to the PSI-LHCI structures of land plant, green alga, and red alga. The binding sites between FCPIs and LHCIs are conserved to some extent. However, our recent study revealed that no orthologous relationship exists among LHCs bound to PSI between primitive red algae and diatoms (Kato et al., 2024). Consequently, we found that the information obtained from structural comparisons alone is extremely limited. To avoid misinterpretation, this study focused on comparing the structures and amino acid sequences of FCPIs between *T. pseudonana* and *C. gracilis*.

(4) Line 104: Despite achieving high resolution, the authors modeled only six lipid densities (the PDB model contains actually 9 lipids, you should correct it in the text). Do you believe this is due to the detergent used for purification? Can you comment on the position, identity, and potential role of the lipids within your model?

There are 6 lipids associated with the PSI core and 3 with FCP, giving rise to a total of 9 lipids. We have described it in our original text (lines 102-104 in the modified manuscript). Additionally, our structure reveals unidentified densities which likely represent lipids; they are modeled as 88 unknown lipids (UNLs). Thus, there are more lipids in the supercomplex. However, we also observed 4 β-DDM molecules (LMT) in the structure, which are used as detergents. Thus, it is possible that some lipids have dissociated and replaced by detergents. Many of the observed lipids are located between subunits, likely contributing to the stabilization of the complex.

(5) Line 111: The global resolution is very high. Why does the unknown protein have such low resolution that it was impossible to model it properly and perform de novo identification from the density map? Is it due to a lower abundance of particles with this subunit bound? Have you tried improving this with 3D classification/ focus refinement /density modification?

The Unknown subunit (UNK) is located peripherally, and its density is significantly lower compared to the neighboring subunits, which may suggest a low abundance. We applied density modification using Topaz for 3D map denoising, but the effect was minimal. As the low abundance of UNK may be the cause, 3D classification and focus refinement also had limited impact.

(6) Figure 2A: It would be useful to show the density map for the subunit together with the model, especially to demonstrate visualization of the long loop.

We added the model and map of Psa29 to Figure S4C in the revised manuscript.

(7) Given the proximity of Psa29 to PsaC, is the protein involved in electron shuttling? If so, could you comment on this? In line 131, you state that Psa29 was not found in other organisms. Can the authors speculate on the potential role of this protein in diatoms?

We have no idea about the function of Psa29 at present. However, Psa29 does not contain any cofactors, indicating no contribution of it to electron transfer reactions. To understand the function of Psa29, a deletion mutant of this gene is required for examining its functional and physiological roles in diatom photosynthesis. To explain this, we added the following sentences to the revised manuscript (lines 129-133):

"However, the functional and physiological roles of Psa29 remain unclear at present. It is evident that Psa29 does not have any pigments, quinones, or metal complexes, suggesting no contribution of Psa29 to electron transfer reactions within PSI. Further mutagenesis studies will be necessary to investigate the role of Psa29 in diatom photosynthesis."

(8) Line 163: "Among the FCPI subunits, only FCPI-1 has BCRs in addition to Fxs and Ddxs (Figure S6A). FCPI-1 is a RedCAP, which belongs to the LHC protein superfamily but is distinct from the LHC protein family (6, 7)." It would be useful if the authors could add the carotenoid model embedded in the cryoEM density map to the figure to show the features that led to modeling BCR instead of other carotenoids. Additionally, it would be helpful to include in the text why RedCAPs differ from LHCIs and their proposed role.

We added the model and map of two BCRs in FCPI-1 (RedCAP) to Figure S4F in the revised manuscript.

Phylogenetic analysis showed that RedCAPs are distinct from the LHC protein family. This has been explained in lines 163-164. Also, the functional and physiological roles of RedCAP remain unclear. To explain this, we added the sentence "; however, the functional and physiological roles of RedCAP remain unclear" to the revised manuscript (lines 164-165).

(9) Line 185: "However, it is unknown (i) whether CgRedCAP is indeed bound to the C. gracilis PSI-FCPI supercomplex and (ii) if a loop structure corresponding to the Q96-T116 loop of TpRedCAP exists in CgRedCAP." Have the authors attempted to model the protein using AlphaFold? If so, are there significant differences? Could you speculate on the absence of RedCAP in C. gracilis? Do you believe it is due to using a different detergent or related to environmental factors?

We did not model CgRedCAP using AlphaFold. Our recent study “Kato et al. 2024” proposed that CgRedCAP binds to the LHCI-1 site in the PSI-FCPI structure based on sequence comparison. There are two types of PSI-FCPI supercomplexes, one having 16 FCPIs and the other having 24 FCPs, from *C. gracilis*. The different antenna sizes may depend on the growth conditions of *C. gracilis* (Nagao et al. 2020). These explanations were already described in the manuscript (lines 243-246).

(10) Line 193: Figure 8 is mentioned before Figures 4-7.

We are sorry for the mistake of Figure number. Figure 8 is Supplementary Figure 8, so that we modified Fig. S8B in the revised manuscript.

(11) Line 223: FCPI-4 interacts only with FCPI-5, primarily through the interaction of Y196/4 with the FCPI-5 backbone. Is this interaction facilitated by other factors such as lipids, carotenoids, or other ligands? Also, FCPI-4 occupies a peculiar position compared to other LHCIs proteins (it is peripheral to FCPI-4 and FCPI-5). Do you believe this could be due to a transient interaction with the complex? Could the presence of this protein be related to the growth conditions experienced by the plant? Are there any literature reports on environmental conditions influencing FCPI arrangements? Including this information in the text would be interesting.

Y196/4 interacts with only backbones by hydrogen-bond interactions; therefore, other cofactors do not contribute to the interactions.

We do not believe that the interaction of FCPI-4 is transient; rather, this binding appears to be stable within the complex. Given that the PSI-FCPI supercomplexes were isolated by anion exchange chromatography, FCPI-4 and FCPI-5 are tightly associated within this complex. However, it is important to note that the expression of diatom FCPI proteins can indeed vary depending on growth conditions, as highlighted in our previous study (Nagao et al., 2020). While the peculiar position of FCPI-4 may not be directly related to transient interactions, environmental conditions could still influence the overall arrangement and expression levels of FCPIs. This information has already been described in the manuscript (lines 243-246).

(12) Given the high resolution of your map, the overall model quality does not seem to match the map quality. Specifically, the clash score (10) and sidechain outliers (3%) are elevated. Could you comment on this? Do you believe it is related to the high number of ligands?

Our structure contains a total of 295 ligands, including cofactors, detergents, and unknown lipids. We believe the high clash score and number of sidechain outliers are due to the large number of ligands present.

(13) Supplementary Figure 2: You should show the 3D classes that were discarded.

According to your comment, we added the 3D classes that were discarded and the sentence "Red boxes highlight selected particles from each 3D classification." to Figure S2 and its legend in the revised manuscript.

(14) Which masks were used for refinement? How were they generated, and which parameters were chosen? This information should be added to the Materials and Methods section. You should show the masks used during classification, for example.

We used a 240 Å spherical mask for refinement and classification, without applying any reference mask as input. To explain this, we added the corresponding sentence to Methods in the revised manuscript (lines 347-348) as follow:

"A 240-Å spherical mask was used during the 3D classification and refinement processes."

(15) Were any extra proteins detected in the early stages of the cryoEM analysis (i.e., 2D classification) that were discarded? Could you visualize the superior oligomeric states of the supercomplex?

In the single-particle analysis, no larger particles than the analyzed complex were detected. The results of 2D classification using a sufficiently large spherical mask with a diameter of 320 Å are shown below.

**Author response image 1. sa2fig1:** 

(16) Have you tried using cryoSPARC for data analysis? If so, could you comment on that?

We did not use cryoSPARC for data analysis.

**Reviewer #2 (Recommendations For The Authors):**
I have some minor comments below to help improve the manuscript. The line numbers below refer to those in the Word version of the manuscript.(1) Figure 1 legend, line 559, "membrane normal"? Panel A and B, structures with the same colors, do they refer to the closely related or interacted parts? For example, the red color for FCP1-1 in A and PsaA in B. If not, the authors may want to clarify it.

The term 'membrane normal' refers to the direction perpendicular to the surface of a membrane. It is a concept frequently used in physics and biology to describe the orientation relative to the membrane's plane.

We do not refer to either the closely related or interacted parts used in Figure 1. According to your comments, the colors of subunits were revised in the revised manuscript.

(2) Line 109-117. "Psa28 is a novel subunit found in the C. gracilis PSI-FCPI structure, and its name follows the nomenclature as suggested previously (31).... After psaZ, the newly identified genes should be named psa27, psa28, etc., and the corresponding proteins are called Psa27, Psa28, etc... Psa28 was also named PsaR in the PSI-FCPI structure of C. gracilis (16)". It is confusing. Was Psa28 named twice, PsaR and Psa28? It would be helpful to add a simple explanation here.

According to your comment, we modified the sentence as follow (lines 117-118):

" However, Xu et al. named the subunit as PsaR in the PSI-FCPI structure of *C. gracilis* "

(3) Line 134, "One of the Car molecules in PsaJ was identified as ZXT103 in the T. pseudonana PSI-FCPI structure but it is BCR112 in the C. gracilis PSI-FCPI structure (15)". Figure S4D mentioned BCR863 but did not mention BCR112. Figure S4C, D, it may need better explanations of the colors and labels, and indicate which parts are from T. pseudonana or C. gracilis.

BCR112 was misnumbered; the correct number is BCR103. In response to your comments, we revised Figure S4C and D by labeling the characteristic pigments in the revised manuscript.

(4) Figure S7, although mentioned in the legend, it would be helpful to label interaction pairs on the figure directly with corresponding colours.

According to your comments, we modified the Figure and legends in the revised manuscript.

(5) Figure 3E, it is better to avoid red/green colours in one figure as some readers may be colour-blind. It would also be helpful to label each FCPI with the same colour as its structure on the figure directly.

According to your comments, we modified Figure 3E in the revised manuscript.

(6) Line 185, "structures similar to the Q96-T116 loop in TpRedCAP found in the present study (Figure 8B).". The authors refer to Figure S8B? I have the same comment for line 186, Figure 8C.

We are sorry for the mistake of Figure number. Figure 8 is Supplementary Figure 8, so we modified it as Fig. S8B in the revised manuscript.

(7) Line 270, "TpLhcq10 cannot bind at the FCPI-2 site". Why not use FCPI-3 for TpLhcq10?

This means that the gene product of TpLhcq10 binds at the FCPI-3 site but not at the other sites such as FCPI-2. To avoid misreading, we modified the sentence as follows:

"TpLhcq10 binds specifically at the FCPI-3 site but not at the other sites such as FCPI-2" (lines 278-279)

**Reviewer #3 (Recommendations For The Authors):**
I have no technical or conceptual suggestions at the current stage.

Thank you.